# Correlated electronic states at domain walls of a Mott-charge-density-wave insulator 1$T$-TaS$_2$

Doohee Cho[1,2], Gyeongcheol Gye[2], Jinwon Lee[1,2], Sung-Hoon Lee [1], Lihai Wang[3,4], Sang-Wook Cheong[3,4] & Han Woong Yeom[1,2]

Domain walls in interacting electronic systems can have distinct localized states, which often govern physical properties and may lead to unprecedented functionalities and novel devices. However, electronic states within domain walls themselves have not been clearly identified and understood for strongly correlated electron systems. Here, we resolve the electronic states localized on domain walls in a Mott-charge-density-wave insulator 1$T$-TaS$_2$ using scanning tunneling spectroscopy. We establish that the domain wall state decomposes into two nonconducting states located at the center of domain walls and edges of domains. Theoretical calculations reveal their atomistic origin as the local reconstruction of domain walls under the strong influence of electron correlation. Our results introduce a concept for the domain wall electronic property, the walls own internal degrees of freedom, which is potentially related to the controllability of domain wall electronic properties.

---

[1] Center for Artificial Low Dimensional Electronic Systems, Institute for Basic Science (IBS), 77 Cheongam-Ro, Pohang 790-784, Korea. [2] Department of Physics, Pohang University of Science and Technology (POSTECH), Pohang 790-784, Korea. [3] Laboratory for Pohang Emergent Materials, Pohang University of Science and Technology (POSTECH), Pohang 790-784, Korea. [4] Rutgers Center for Emergent Materials and Department of Physics and Astronomy, Rutgers University, Piscataway, NJ 08854, USA. Correspondence and requests for materials should be addressed to H.W.Y.(email: yeom@postech.ac.kr)

Conductive domain walls and their functionalities[1] have been reported in multiferroic insulators[2–4], magnetic insulators[5,6], Mott insulators[7], and layered charge density wave (CDW) materials[8–11]. CDW with periodic modulations of charge/lattice at low temperature and their domain walls are widely observed in metallic layered transition metal dichalcogenides[12]. Among them, $1T$-$TaS_2$ shows a series of CDW transitions coupled to substantial electron-electron interaction. Especially, a metal-insulator transition occurs below 190 K through a transition from a nearly commensurate to a commensurate CDW state[13]. The insulating ground state is realized by electron correlation on narrow electron bands formed by the commensurate CDW lattice[14]. Recent experimental results have demonstrated that the correlated insulating ground state can be transformed into various quasimetallic and superconducting CDW states by external and internal control parameters, such as pressure[8], optical (electrical) excitation[15–20], and chemical doping[21–23]. These conductivity switchings are intrinsically very fast, which may make possible novel ultrafast devices based on correlated electrons.

In the metallic excited states, CDW domain walls are common objects and considered as the origin of the metallic property. They have been suggested as highly conducting channels themselves[8,15,17] and/or to screen electron correlation within Mott-CDW domains[21]. Moreover, the emerging superconductivity out of the Mott-CDW insulating phase has been considered being directly related to conducting domain walls[8,9,24]. However, despite such long discussion, the electronic states of domain walls have not yet been clarified spectroscopically. Here, we show that the domain walls in $1T$-$TaS_2$ have two well-confined and non-metallic in-gap states above Fermi level ($E_F$) using scanning tunneling microscopy and spectroscopy (STM and STS). They are located on the domain wall center and edges of neighboring domains, respectively. The theoretical

calculations strongly suggest the substantial correlation effect in forming spatially decomposed and non-metallic domain wall states.

## Results

**Domain walls in the insulating CDW state**. Figure 1a shows a typical STM image of the commensurate Mott-CDW state in $1T$-$TaS_2$ at 4.3 K. The unit layer of $1T$-$TaS_2$ consists of a Ta layer sandwiched by two S layers with each Ta atom coordinated octahedrally by S atoms. Unpaired $5d$ electrons of Ta form a half-filled metallic band which is unstable against the CDW formation driven by strong electron-phonon coupling. Ta atoms undergo a reconstruction into a so-called David-star unit cell which is composed of 13 Ta atoms; the 12 outer atoms pair up and shift toward the central one[25], where a unpaired $5d$ electron is left over[14,26]. In the commensurate CDW phase, David-star unit cells (green lines in Fig. 1a) exhibit a regular triangular lattice with a period of 12.1 Å and its STM image is dominated by protrusions representing unpaired electrons of central Ta atoms[27]. The onsite Coulomb repulsion drives these unpaired electrons into a Mott insulating state, which form otherwise a narrow metallic band. In this way, the insulating state is driven by a cooperative interplay between the CDW and the electron-electron interaction[28].

In spite of the clear commensurability and the long range order of the undoped low temperature phase, there exist one-dimensional intrinsic defects, domain walls, across which the phase of CDW is abruptly shifted (see Fig. 1b and Supplementary Fig. 1). Since there can be 13 atoms within a CDW unit cell, 12 antiphase domain wall configurations are possible as indexed in Fig. 1a[19]. The configuration of a domain wall can be easily identified by the phase shift (black arrows in Fig. 2a, b) of CDW protrusions of neighboring domains (see Supplementary Fig. 2). Only few cases among 12 configurations are observed including

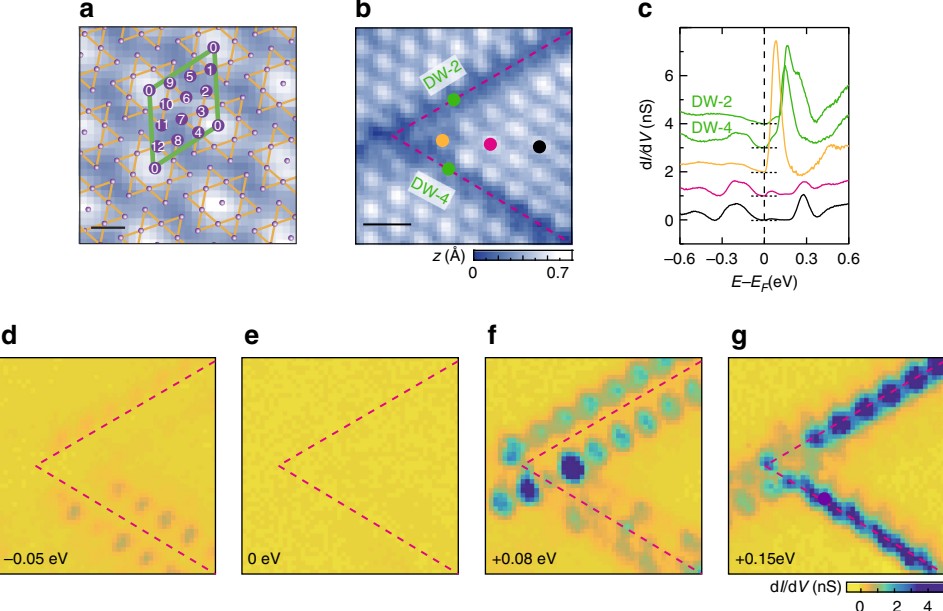

**Fig. 1** Domain walls of 13-fold degenerate CDW reconstruction in $1T$-$TaS_2$. **a** Constant current STM image of the commensurate CDW with a schematic view of the David-star (orange lines) reconstruction (sample bias voltage $V_s = -1.20$ V, tunneling current $I_t = 100$ pA, and scan size $L^2 = 3.4 \times 3.4$ nm$^2$). The violet spheres display the Ta atoms encapsulated by the S layers. The numbers in the CDW unit cell (a green parallelogram) indicate possible center positions of the 13-fold degenerate David stars. The indices are given by the conventional sequence[19]. Scale bar, 0.5 nm. **b** A STM image of the junction composed from domain walls (dashed lines) ($V_s = -1.20$ V, $I_t = 100$ pA, and $L^2 = 9.0 \times 9.0$ nm$^2$). Scale bar, 2 nm. **c** d$I$/d$V$ spectra acquired at the position marked by same colored dots in **b**. Each curves are equally shifted in intensity for clarity. **d**–**g** Spatial distributions of the domain wall and edge states at the energies given in the figures

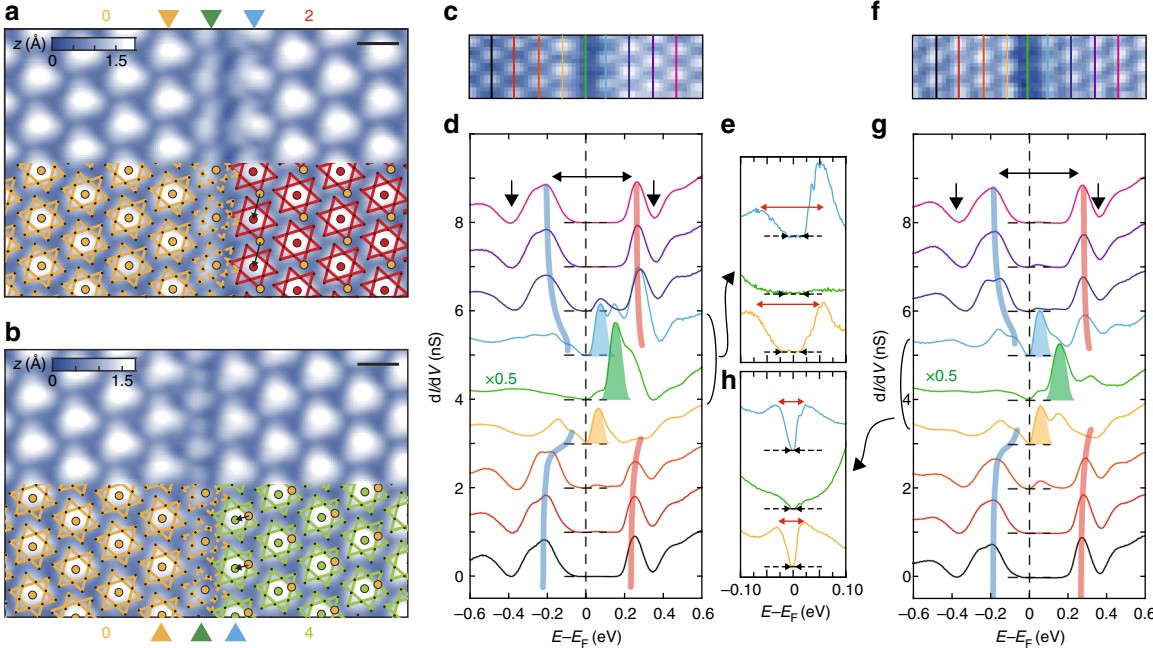

**Fig. 2** Atomic and electronic structure of domain walls and edges. **a, b** Constant current STM images ($V_s = -0.5$ V, $I_t = 500$ pA, and $L^2 = 9.7 \times 6.4$ nm$^2$) of the second and the fourth domain wall, superimposed with the schematic David-star superlattice. The indices for each domains are indicated in Fig. 1a. The black arrows represent the phase shift between neighboring domains. The center and edges of domain walls are marked by *green and yellow (blue) triangles*, respectively. *Scale bar*, 1 nm. **c, f** The STM images ($V_s = -1.2$ V, $I_t = 100$ pA) are simultaneously acquired with the STS measurements. **d, g** Spatially resolved $dI/dV$ spectra cross the domain walls. Each spectrum is averaged in the region marked by same colored lines in the STM images **c, f**. The evolution of the incoherent peaks in the domain is displayed by the semitransparent *blue* and *red* curves. The *double-headed black arrows* display the gap size of domains. The *single-headed black arrows* indicate the subband splitting. The in-gaps states are highlighted by the filled gaussian-shaped peaks. **e, h** High-resolution STS spectra at the domain wall and edges. The *double-headed black* and *red arrows* indicate the zero conductance region and the peak-to-peak splitting at the domain edges, respectively. Spectra are equally shifted in intensity for clarity

the most popular ones shown in Fig. 2, probably because of the energetics and/or kinetics of the domain wall formation[19,29]. The present work covers two most popular and straight domain walls (indexed as the second (upper) and the fourth (lower) one following the atomic indices of Fig. 1a) which were found to be stable within the experimental timescale. We mainly focus on the second domain wall which is the majority species[18,19].

**Localized in-gap states of the domain wall and edges.** Electronic states localized on domain walls are revealed by acquiring STS spectra around domain walls with a high spatial resolution. The previous STS work showed only enhanced spectral weight around the band gap region without observing any distinct electronic states[19]. In the present work, for example, STS spectra were measured at each pixel of the STM image containing two domain walls of different but most popular types (Fig. 1b). Within the Mott gap region of $-0.1$ to $+0.2$ eV around $E_F$, one can observe two pronounced localized electronic states. A strong spectral feature appears along the center of domain walls in the empty state at $+0.15$ eV (two *green curves* in Fig. 1c, g) and another one also in the empty state at $+0.08$ eV (a *orange curve* in Fig. 1c, f) but slightly away from the center of domain walls. The latter is localized along the edge of neighboring domains and a similar but weak feature also appears in the filled state at $-0.05$ eV (Fig. 1d). Note that we do not observe any distinct spectral weight at $E_F$ which is essential for domain-wall-originated metallicity or superconductivity (Fig. 1e). The present result is consistent with our recent work for nearly commensurate domain wall networks[18]. The localized in-gap state above $E_F$ was also observed on zigzag domain walls (see Supplementary Fig. 3).

The $dI/dV$ maps at the energies of the in-gap states show charge modulations along domain walls and edges. Their periodicities are the same as that of the David-star reconstruction. These domain wall and edge states exhibit specific phase relations determined by the phase difference between neighboring domains across the domain wall (see Supplementary Figs. 2 and 4). This implies that the domain wall reconstruction is strongly influenced by the periodic potential imposed by the CDW reconstruction in neighboring domains[30].

Gapped nature of domain walls and their localized electronic states become more clear in a closer look of point-by-point STS spectra. Figure 2a, b show well-resolved STM images of the second and fourth domain walls, which are the same types as those of Fig. 1b. Within domain walls, David-star CDW units cannot be formed completely and the incomplete David-stars reconstruct into pairs of small and large protrusions along domain walls. This will be discussed in more detail below. A series of $dI/dV$ spectra across domain walls are shown in Fig. 2d, g. Away from domain walls, the $dI/dV$ curves reproduce the energy gap of the Mott-CDW state; two Hubbard states construct a Mott gap of $0.44 \pm 0.02$ eV. The subband splittings (single-headed *black arrows* in Fig. 2d, g) are footprints of the formation of CDW and the David-star reconstruction[26,28]. Upon approaching the domain wall, CDW protrusions and STS spectra do not show a noticeable spatial variation, except for a tiny band bending until they reach the last CDW unit cells of domains, which we call domain edges. As discussed above, at domain edges in both sides of the domain wall, an in-gap state emerges at $+0.07$ eV. Then, at the domain wall center, the Hubbard states are replaced by a strong spectral feature of $+0.15$ eV. Note that the domain wall itself is not metallic at all as also shown in the spatial

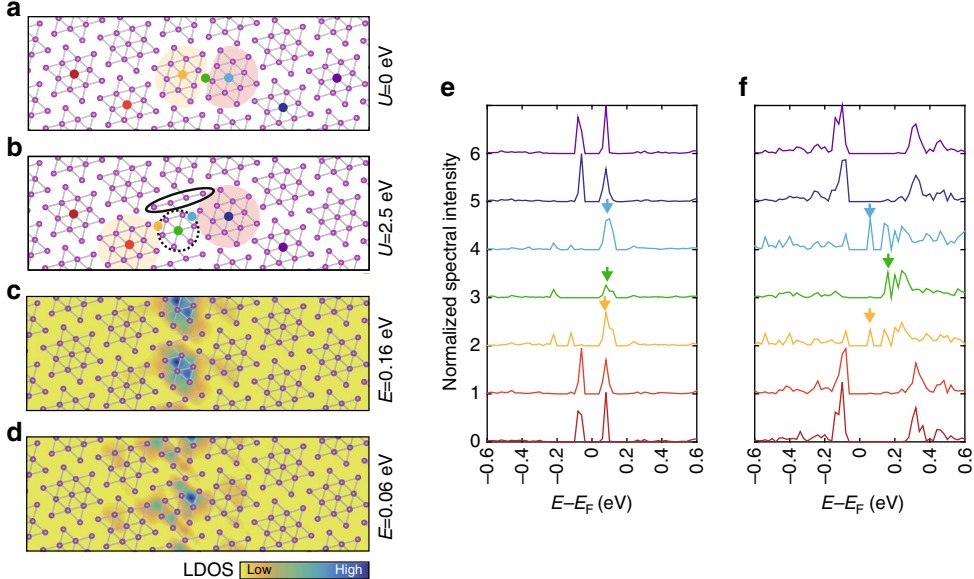

**Fig. 3** Theoretical calculations of $U$-dependent domain wall and edge reconstruction. **a, b** Atomic structure of the second domain walls at $U = 0$ and 2.50 eV. Ta atoms are indicated by *violet spheres*. *Gray lines* indicate the bonding between the nearest neighbors Ta atoms with shorter distances than the primitive unit vector of ×1 structure. **c, d** Calculated spatial distribution of the domain wall and edge states. **e, f** LDOS spectra are acquired at the position marked by colored spheres in **a, b**. The *small arrows* indicate the in-gap states at domain wall and edges. All spectra are normalized by their maximum value and equally shifted for clarity

maps at $E_F$ (Fig. 1e). While the edges of domains have higher spectral weight near $E_F$, the band gap on the domain edge can be unambiguously defined by the edge of the lower Hubbard band and the new spectral feature above $E_F$. Their peak-to-peak splitting are 120 and 50 meV on the second and fourth domain wall, respectively (*red double headed arrows* in the Fig. 2e, h). Due to thermal and instrumental broadenings, these spectral features leave decaying intensities toward $E_F$, which obscure the band gap. However, our higher resolution STS measurements (see Methods) clearly show the zero conductance region at $E_F$ (see Fig. 2e, h and Supplementary Fig. 5), which makes the existence of the band gap unambiguous.

**Correlation-dependent domain wall and edge reconstructions.** In order to elucidate the origin of the domain wall and edge states, first principles calculations were performed. We used a domain supercell in a monolayer $1T$-TaS$_2$ in which a single domain is composed of five columns of David stars. Each domain is shifted to construct domain wall of the second type between neighboring domains. The equilibrium structure after a full relaxation of atomic positions is shown in Fig. 3a. Here we use the usual density functional theory framework without putting any extra electron correlation, which substantially underestimates the Mott-CDW band gap at $E_F$ (Fig. 3e)[31]. The two overlapped David-star columns in the domain wall region simply split into two symmetric rows of incomplete David stars with 12 Ta atoms on each. This produces a unique electronic state on the domain wall at the energy of conduction band minimum (*arrows* in Fig. 3e). This domain wall structure and its electronic state deviate qualitatively from what were observed. We then include the extra electron correlation effect with a finite $U$ value within the GGA + $U$ scheme. The $U$ value is tuned up to 2.5 eV in order to properly reproduce the experimentally observed Mott gap of the CDW domain (Fig. 3f). With the enhanced electron correlation, the domain wall reconstruction drastically changes; it becomes much narrower with 11 Ta atoms, which split

longitudinally into centered hexagons and linear tetramers (see Supplementary Fig. 6). The atomic lattice of the domain wall is resolved in our high-resolution STM topography while small lattice distortions due to the reconstruction within are hardly detected[19] (see Supplementary Fig. 7). However, strong charge modulations relevant with the atomic reconstruction are shown clearly in our STM and STS results. The hexagon on the domain wall center exhibits an electronic state at a slightly lower energy than the upper Hubbard band (a *green arrow* in Fig. 3f). The other states at a lower energy closer to $E_F$ (a *blue and orange arrow* in Fig. 3f) emerge at the edges of hexagons and extend to neighboring David-stars units, that is, to domain edges (Fig. 3d). The fourth domain wall has also the spatially decomposed in-gap states and a smaller gap feature are reproduced in our calculations at $U = 2.5$ eV (see Supplementary Fig. 4). This result strongly suggests the crucial role of electron correlation in the formation of the domain wall electronic state.

We note that there are still discrepancies between the calculation and the measurement, especially in the spatial distribution of edge states (see Supplementary Fig. 4). The maxima of calculated edge states (Fig. 3d) are located closer to the narrow domain wall than those observed while the tendency to spread toward edges of domains is consistent. We suspect that this discrepancy may be related to the interlayer coupling discussed in the previous studies[18,19,31,32], which is not included at all in our calculations.

## Discussion

The electronic state(s) within the band gap observed here is in line with localized states suggested in sharp phase kinks of order parameters in charge or spin ordered insulators[2,6,33,34]. Our STM and STS results disclose that the domain wall is not a metallic channel developed by the suppression of the CDW order, which have been assumed by many previous studies[8,15,17,24]. On the other hand, a few other studies suggested that free carriers from domain wall electronic states screen the electron correlation

in neighboring CDW domains[21]. This idea was critically tested in the present work with a clearly negative answer. The domain wall junction shown in Fig. 1b is a good testbed for the screening effect since the CDW domain in between the two domain walls would have an increasingly larger effect of the screening, if any, when approaching closer to the junction point. However, our STS data do not show any substantial change of electronic states from that of the Mott-CDW gap structure as approaching the junction (Fig. 1c), except for the domain wall and edge states (the *yellow spectrum* in Fig. 1c). This result clearly rules out the screening scenario to explain the metallicity of the textured CDW states. Of course, the metallic state can be driven by developing a random disorder potential in Mott insulators and decreasing the CDW lateral or vertical ordering[18]. However, the single isolated and straight domain wall as discussed here is not related to such a global disorder to melt the correlation gap[35,36].

In summary, we discover distinct electronic states of domain walls of a symmetry-broken correlated insulator for the first time. The domain walls of the Mott-CDW phase of $1T$-TaS$_2$ have well-localized non-metallic in-gap states along the center of domain walls and the edges of neighboring domains. Not only the lack of free carriers but also the lack of any substantial screening or doping effects by domain walls are disclosed unambiguously. These results request to rewrite most of the current scenarios on the metallic and superconducting phases emerging from this correlated insulator. The origin of the split non-metallic domain wall states is clearly understood as due to the reconstruction within domain walls composed of multiple atoms and electrons under the substantial influence of electron correlation. That is, the internal structural and electronic degrees of freedom within domain walls are indicated to be very important. The internal structural degrees of freedom of domain walls were recently noticed for one[34] and two-dimensional[30] systems, but to the best of our knowledge, the substantial electron correlation effect within a domain wall has not been observed before. These internal degrees of freedom might be exploited to provide controllability of domain wall electronic properties for the functionalization of complex materials with domain walls.

## Methods

**Experiments**. The STM and STS measurements have been carried out with a commercial STM (SPECS) at 4.3 K. Pt-Ir wires were used for STM tips. All STM images were acquired with the constant current mode with bias voltage ($V_s$) applied to the sample. The standard lock-in technique with voltage modulation $V_m = 10$ mV and frequency $f = 1$ kHz has been adapted for acquiring d$I$/d$V$ spectra. The high-resolution spectra were obtained with $V_m = 2$ mV. A single crystal of $1T$-TaS$_2$ was prepared by iodine vapor transport method, as described in ref. [18]. The samples were cleaved at room temperature and quickly transferred to the precooled STM head.

**Theoretical calculations**. Our calculations were carried out using density functional theory (DFT) with the projector augmented wave method, Perdew-Burke-Ernzerhof exchange and correlation functional and GGA + $U$ as implemented in the Vienna *ab initio* simulation package. To avoid interactions between the supercell, about 10 Å vacuum is inserted in vertical direction. The plane wave cut-off energy is set to 260.0 eV and the Monkhorst-Pack $k$-point mesh is $1 \times 2 \times 1$. All the atoms are relaxed until forces on the atoms are <0.02 eV/Å.

**Data availability**. The authors declare that the data supporting the findings of this study are available within the article and its Supplementary Information.

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

## Acknowledgements

This work was supported by the Institute for Basic Science (Grant No. IBS-R014-D1). L.W. and S.-W.C. are partially supported by the Max Planck POSTECH/KOREA Research Initiative Program (Grant No. 2011-0031558) through NRF of Korea funded by MEST. S.-W.C. is also supported by the Gordon and Betty Moore Foundations EPiQS Initiative through Grant GBMF4413 to the Rutgers Center for Emergent Materials.

## Author contributions

D.C. and H.W.Y. conceived the research idea and plan. D.C. and J.L. performed the STM/STS measurements. L.W. and S.-W.C. grew the single crystals. S.-H.L. and G.G. conducted the DFT calculations. D.C. and H.W.Y. prepared the manuscript with the comments of all other authors.

## Additional information

**Competing interests:** The authors declare no competing financial interests.

