## [Peer Review File · Nature Communications]

Reviewers' comments:

Reviewer #1 (Remarks to the Author):

Review of "Correlated electronic states at domain walls of a Mott-charge-density-wave insulator 1T-TaS₂" by Doohee Cho, Gyeongcheol Gye, Sung-Hoon Lee, Choongjae Won, Sang-Wook Cheong, and Han Woong Yeom

The manuscript "Correlated electronic states at domain walls of a Mott-charge-density-wave insulator 1T-TaS₂" by Cho et al. describes a scanning tunneling microscopy/spectroscopy study on gap states, appearing at grain boundaries in 1T-TaS₂. The authors show that the gap states they find at the boundary are localized via differential conductive mapping and argue that the boundaries are not conducting and that theoretical calculations reveal that the edge state is a strongly correlated electron system.

In principle, I agree, that revealing and understanding the electronic structure of edge or boundary states in any materials that undergoes a quantum phase transition such as Mott-CDW insulators is of great interest, due to the fundamental potential of finding new material properties due to competing symmetry breaks in these systems. Hence, I believe that results such as these presented here are worthwhile to be published in nature communication. However, I am also not convinced about the interpretation of the here presented results nor some of the resulting claims, and suggest some major revisions before publishing these results. Here are the main point I believe should be addressed:

1. I have problems to accept the claim that the boundary and hence the boundary electronic states are not metallic: The spectra sure looks like a metallic STS spectra. The dI/dV map shows clearly a state at -0.05 eV and at $+0.08$ eV, which is very close to the Fermi level, closer then the energetic resolution of the measurements due to the 10mV modulation amplitude for the lock in amplifier used for these experiments. A featureless dI/dV map at 0 V does not prove an insulating state since there is no tunneling current to be had in the first place. To claim that there is indeed a small gap remaining I strongly suggest to take tunneling spectra at the grain boundary only from $+0.1V$ to $-0.1V$ using a lock in amplitude of not more then 2mV to reveal with high energetic resolution any potential gap-like features around the Fermi level. Subsequently, repeating the differential maps at ± 0.02 eV would indeed prove that there is a gap of at least 4meV that with no conductance. A clear state at 0.05eV I would interpret at metallic.
2. An interesting feature that is hardly discussed in the manuscript, which made me think this could have some signes of a correlates system is that the conductance maps at 0.08 eV and 0.15 eV have interesting periodicities and at first sight seem to be out of phase at first glance. Here I would suggest to investigate if the wavelength of the periodic signal at $-0.05eV$, 0.08 eV and 0.15 eV are the same, is the periodicity related to the CDW induced formation of the david stars or independent of it, and if independent, can you detect a reproducible phase shift between the boundary induced periodic states at these 3 energies.
3. The atomistic structure of the grain boundaries come for an extrapolation of the david star structure. However, energetics can be very complicated at grain boundaries, where the actual atomic structure can be substantially altered compared to the extrapolation from the atomic structure close to the boundary condition. Getting the atomistic

structure correct is key to plug in the structure into a Hubbard model. Here I suggest one of two approaches: 1. The most ideal is using in parallel a non contact AFM to STM, which provides an exact atomic structure. 2. Since most groups do not have the ability to do both yet, some high resolution STM images, which show the atomic structure of the david star formation close to the boundary condition might give at least an educated guess of the boundary morphology. I give the authors however credit to have optimized the domain boundary structure via DFT which gives some indication that the structure plugged into the Hubbard model is energetically the favorable one. However, DFT is not exactly famous for getting energetics of the electronic structure or the relaxation of the atomic structure at defects or grain boundaries right.

4. I find the claim that the theoretical the experimental states match well somewhat deceptive. Going back and fourth between Fig 2 and 3 I see many peaks in Fig. 3 above and below the Fermi level that I cannot identify in the experimental results. My suggestion here is to take again an energetically high resolution dI/dV spectra and lie it over a theoretical spectra that mimics the LDOS a few Angstroms above the grain boundary based on the calculated gap states to have a clear comparison. In the current form the experimental and theoretical agreement are not very evident to me.
5. While the argument of a correlated system for the actual Mott-CDW insulator is obviously correct, it is not obvious to me that this claim can be made for the actual grain boundary states. It is not obvious to me how the argument of a breakdown of the Mott insulator into a correlated metal by doping or CDW disorder from the same group can be applied and hence argued here. Since the manuscript makes the claim here that the grain boundary states are correlated, which would be indeed an important finding, I believe that it needs to be more clearly explained to be able to make the claim. It is good possible that I might not have enough insight to fully understand and appreciate the here presented argument, while an expert in correlated electron and topological systems would follow. However, since the authors address a more general audience in nature communications and it is a key argument for the claim of this paper I suggest to present the argument of correlation clear enough so that a audience with a solid state physics background can follow.

Reviewer #2 (Remarks to the Author):

The authors report on electronic states near one type of domain wall in 1T-TaS₂, a Mott-CDW insulator. This study combines STM/STS measurements with DFT calculations and highlights the presence of localized electronic state at and near a domain wall measure on the surface of the material. Domain walls in 1T-TaS₂, at which the phase of the CDW shifts abruptly, have been suggested to provide conductive channels and to play a role in emergent superconductivity. The work presented here directly probes electronic states near domain walls and shows that they themselves are non-conducting. There are, however, localized states present at the center and near the domain wall which are distinct. This work is timely and presents an important contribution which I recommend for publication in Nature Communication once the following points are addressed:

To further strengthen the paper and to improve on generality of the work I suggest the authors add a discussion on other types of domain walls such as the zigzag one shown in the SI. Are these domain wall conductive from STS?

The DFT results presented in Fig. 3 are only showing DW-2 domain walls. Please also include results for DW-4 (in SI). I understand that the experimental data shows more pronounced electronic states for DW-2 domain walls, however, it would be important to verify the trends for DW-4 as well.

In the figure 3 caption please specify the energy for (c) and (d).

When comparing the STS and DFT maps of the domain wall and edge states it appears that they don't quite match (by rough alignment using Fig. 1b as reference). Specifically, the maxima of the domain wall states (1g) are not located right between the max in 1b, i.e. the centers of the stars-of-David. This might suggest that the reconstruction predicted by DFT is not quite accurate. Please include a figure that shows the position of the domain wall states with respect to the stars-of-David in the neighboring domains (for the experimental data) and discuss the discrepancy with the DTF results.

There are minor wording errors/typos that I assume will be taken care of in the next iteration.

Reply to reviewer-1's comments:

In principle, I agree, that revealing and understanding the electronic structure of edge or boundary states in any materials that undergoes a quantum phase transition such as Mott-CDW insulators is of great interest, due to the fundamental potential of finding new material properties due to competing symmetry breaks in these systems. Hence, I believe that results such as these presented here are worthwhile to be published in nature communication. However, I am also not convinced about the interpretation of the here presented results nor some of the resulting claims, and suggest some major revisions before publishing these results. Here are the main point I believe should be addressed:

Comment 1-1 : I have problems to accept the claim that the boundary and hence the boundary electronic states are not metallic: The spectra sure looks like a metallic STS spectra. The dI/dV map shows clearly a state at -0.05 eV and at $+0.08$ eV, which is very close to the Fermi level, closer than the energetic resolution of the measurements due to the 10mV modulation amplitude for the lock in amplifier used for these experiments. A featureless dI/dV map at 0 V does not prove an insulating state since there is no tunneling current to be had in the first place. To claim that there is indeed a small gap remaining I strongly suggest to take tunneling spectra at the grain boundary only from $+0.1$ V to -0.1 V using a lock in amplitude of not more than 2mV to reveal with high energetic resolution any potential gap-like features around the Fermi level. Subsequently, repeating the differential maps at ± 0.02 eV would indeed prove that there is a gap of at least 4meV that with no conductance. A clear state at 0.05eV I would interpret as metallic.

Reply 1-1 : We appreciate that the reviewer pointed out this issue. However, we have to say that the dI/dV intensity is indeed strictly zero for the domain wall center at least and its main spectral feature is well away from the Fermi level at $+0.13$ eV. This can be clearly seen in our data in revised Supplementary Fig. 4 and insets of Fig. 2 in the main text. Note that even if tunneling current (I) is zero at $V=0$, a metallic system would have the finite dI/dV value proportional to metallic LDOS [or the finite slope of $I(V)$ curves]. Thus, zero dI/dV value at $V=0$ is a clear indication of the non-metallic behavior.

For the domain edges, the situation is not as clear as the domain wall with their spectral features much closer to the Fermi level and this would be the case pointed out by the reviewer. However, please also note that a metallic system would not

normally exhibit a strong dip feature as observed here. Anyway, we may need to be more careful here. The energy resolution of STS measurements is 25 meV as determined as $dE = \sqrt{(3.3k_B T)^2 + (2.5eV_{mod})^2}$, where k_B is the Boltzmann's constant, and V_{mod} the lock-in modulation. Thus, the thermal broadening and the lock-in modulations can make non-zero dI/dV intensity within the energy gap. Even in this case, the reviewer would agree that this broadening does not mean an intrinsically metallic state. For example, in order to reproduce such measured spectra, we can apply Gaussian convolution (25 meV FWHM) to the calculated ones with clear energy gaps at the Fermi level. The broaden spectra are reasonably consistent with the measured ones (see below).

In order to further confirm our claim, as requested by the reviewer, we performed higher resolution dI/dV measurements with smaller lock-in modulation (2 meV) and at 4.3 K. The results show the finite size gaps with zero dI/dV more clearly (see below). We included these new STS results in Fig. 2 and added brief comments in the main text (see the summary of the changes). We thus believe that the insulating character of domain walls here is not ambiguous at all.

Comment 1-2: An interesting feature that is hardly discussed in the manuscript, which made me think this could have some signs of a correlated system is that the conductance maps at 0.08 eV and 0.15 eV have interesting periodicities and at first sight seem to be out of phase at first glance. Here I would suggest to investigate if the wavelength of the periodic signal at -0.05 eV, 0.08 eV and 0.15 eV are the same, is the periodicity related to the CDW induced formation of the david stars or independent of it, and if independent, can you detect a reproducible phase shift between the boundary induced periodic states at these 3 energies.

Reply 1-2 : The spatial distributions of the in-gap states at -0.05, +0.08 and +0.15 eV along the domain wall have a periodicity of $\sqrt{13}a_0$ which is same with the CDW lattice constant and their phases are all well consistent with those of neighboring CDW. The domain wall and left edge state (right edge state) follow the CDW periodicity of the left domain (right domain) (see below).

Since two neighboring domains have a phase shift of $(6/13) \times 2\pi \sim \pi$ across the 2nd domain wall along the wall direction, the domain wall state seems to have out-of-phase and in-phase relation with the left and the right edge, respectively (see below). This means that the CDW periodicity along the domain wall direction is well

preserved as shown in our calculations. The filled (-0.05 eV) and empty (+0.08 eV) states at the domain edges for the 2nd domain wall do not show the out-of-phase relation expected in the 1-D CDW systems (**Barja, S. *et al.* Nature Physics 12, 751-756 (2016)**). This point is briefly mentioned in the revised manuscript.

Comment 1-3 : The atomistic structure of the grain boundaries come for an extrapolation of the david star structure. However, energetics can be very complicated at grain boundaries, where the actual atomic structure can be substantially altered compared to the extrapolation from the atomic structure close to the boundary condition. Getting the atomistic structure correct is key to plug in the structure into a Hubbard model. Here I suggest one of two approaches: 1. The most ideal is using in parallel a non-contact AFM to STM, which provides an exact atomic structure. 2. Since most groups do not have the ability to do both yet, some high resolution STM images, which show the atomic structure of the david star formation close to the boundary condition might give at least an educated guess of the boundary morphology. I give the authors however credit to have optimized the domain boundary structure via DFT which gives some indication that the structure plugged into the Hubbard model is energetically the favorable one. However, DFT is not exactly famous for getting energetics of the electronic structure or the relaxation of the atomic structure at defects or grain boundaries right.

Reply 1-3 : We appreciate that the reviewer pointed out this issue. The domain boundaries discussed here are not grain boundary or any other structural line defects but phase mismatches between two CDW domains. Their atomic structures in the first order is, thus, very simple since the whole crystalline lattice is preserved. The phase mismatches of neighboring domains mean the phases of CDW or periodic lattice distortions are mismatched with the underlying lattice preserved. The small lattice distortions for the CDW or its phase mismatch are usually not so well resolved even in atomic resolution STM images nor AFM (see the size of atomic displacements shown in Fig. 6 in Supplementary information). This insensitivity is even worse in the present case since the STM (or AFM) probes not Ta atoms involved directly in CDW but top layer S atoms. The preserved atomic lattice itself was imaged in the previous STM work (**Ma, L. *et al.* Nature Communications 7, 10956 (2016), shown below**) in the metallic incommensurate CDW phase at a very low bias and we also confirmed it. Due to the background metallic phase, this case corresponds to the case of $U=0$ in the present calculation (Fig. 6 in Supplementary information). In clear contrast to this, the same domain boundary within the present

commensurate CDW phase with a band gap has a different reconstruction. The atomic resolution STM image for a DW2 domain in the present case is shown below and attached in Supplementary information too (Fig. 7) as requested by the reviewer. It is clear that the atomic lattice itself is preserved within the domain wall and the David star unit cells persist up to the edges of neighboring domains as our model suggests. Even in this case the small displacements within the domain boundary cannot be easily noticed. As discussed fully in our manuscript, the lattice reconstructions are much more sensitively reflected in LDOS.

Ma, L. *et al.* Nat. Commun. 7, 10956 (2016).

Comment 1-4 : I find the claim that the theoretical the experimental states match well somewhat deceptive. Going back and fourth between Fig 2 and 3 I see many peaks in Fig. 3 above and below the Fermi level that I cannot identify in the experimental results. My suggestion here is to take again an energetically high

resolution dI/dV spectra and lie it over a theoretical spectra that mimics the LDOS a few Angstroms above the grain boundary based on the calculated gap states to have a clear comparison. In the current form the experimental and theoretical agreement are not very evident to me.

Reply 1-4 : We appreciate the reviewer's comment to improve our presentation of the agreement between the calculation and the experiment. As suggested by the reviewer and also the other reviewer, we added a supplementary figure in the revised version to show the agreement or discrepancy in a more clear and detailed way. In the detailed comparison of the dI/dV spectra with the theoretical LDOS, one need to put the energy broadening mentioned above. As shown below, one can see that most of the theoretical and experimental data exhibit excellent agreements except for spatial distributions of the domain edge states of DW2. This remaining discrepancy is made more clear in the LDOS (dI/dV) map comparisons shown below.

As for the comparison of the dI/dV maps, the agreement is also quite good for the domain wall electronic state at 0.15 eV but that for the position of domain edge states slightly deviates as already mentioned (see below). However, the main feature we emphasize here is that this state is extended into the edge unit cells of the two neighboring domains as observed in the experiment. More importantly, these states,

the domain wall states at the center of the domain wall and the edge state extending toward the neighboring domain edges, are qualitatively different from the case with no electron correlation. The partial disagreement for the domain edges might be related with interlayer coupling, which was not considered at all in the calculation. The above points are discussed explicitly in the revised manuscript.

Comment 1-5 : While the argument of a correlated system for the actual Mott-CDW insulator is obviously correct, it is not obvious to me that this claim can be made for the actual grain boundary states. It is not obvious to me how the argument of a breakdown of the Mott insulator into a correlated metal by doping or CDW disorder from the same group can be applied and hence argued here. Since the manuscript makes the claim here that the grain boundary states are correlated, which would be indeed an important finding, I believe that it needs to be more clearly explained to be able to make the claim. It is good possible that I might not have enough insight to fully understand and appreciate the here presented argument, while an expert in correlated electron and topological systems would follow. However, since the authors address a more general audience in nature communications and it is a key argument for the claim of this paper I suggest to present the argument of correlation clear enough so that a audience with a solid state physics background can follow.

Reply 1-4 : We appreciate that the reviewer pointed out that our claim can be more carefully stated, especially, in connection with the disorder or doping by the domain walls. As clearly shown in our spectra, there is little sign of charge transfer or doping by the domain wall. There is only very small band bending near the domain wall. Thus the doping-induced metallization is not relevant here.

The disorder effect is also not relevant either since the single straight domain wall does not induce any global disorder, or, in other words, randomness into the system. The isolated domain walls are not considered to provide enough disorder potential to fully melt the correlation gap (**Lahoud, E. *et al.* Physical Review Letters 112 206402 (2014), Chiesa, S. Physical Review Letters 101 086401 (2008)**). For the disorder effect, we need a certain density of disorder, here multiple irregular domain walls. Reflecting this, the correlation gap is preserved up to the close proximity to the domain wall. This is a similar case to a single neutral point defect. This case can be easily calculated as performed here in contrast to the random disorder case. The disorder in a single domain wall may be introduced with a non-straight and meandering domain wall, which were excluded in the present work.

Imada, M., *et. al.* Rev. Mod. Phys **70**, 1039 (1998). Lahoud, E. *et al.* Phys. Rev. Lett. **112**, 206402 (2014).

The electron correlation effect here is confirmed by the LDA+U (or similarly GGA+U) band structure calculation, which is one of the most well established methods to disclose such correlation effect in a given system. We briefly discussed this in the revised manuscript and provided a few well known references. We discussed this issue in the discussion section in the revised version.

Reply to reviewer-2's comments:

The authors report on electronic states near one type of domain wall in 1T-TaS₂, a Mott-CDW insulator. This study combines STM/STS measurements with DFT calculations and highlights the presence of localized electronic state at and near a domain wall measure on the surface of the material. Domain walls in 1T-TaS₂, at which the phase of the CDW shifts abruptly, have been suggested to provide conductive channels and to play a role in emergent superconductivity. The work presented here directly probes electronic states near domain walls and shows that they themselves are non-conducting. There are, however, localized states present at the center and near the domain wall which are distinct. **This work is timely and presents an important contribution which I recommend for publication in Nature Communication once the following points are addressed.**

Comment 2-1 : To further strengthen the paper and to improve on generality of the work I suggest the authors add a discussion on other types of domain walls such as the zigzag one shown in the SI. Are these domain wall conductive from STS?

Reply 2-1 : We appreciate the reviewer's comment on this point. The zigzag domain wall is not conductive as judged from STS data. It is not commonly observed in this system and not observed in the metallic textured CDW states. This is the reason why we focused only on straight domain walls, especially DW2 (most common domain wall in the metallic textured CDW states).

Comment 2-2 : The DFT results presented in Fig. 3 are only showing DW-2 domain walls. Please also include results for DW-4 (in SI). I understand that the experimental data shows more pronounced electronic states for DW-2 domain walls, however, it would be important to verify the trends for DW-4 as well.

Reply 2-2 : We appreciate the reviewer’s comment on this point. Since DW-2 is most commonly observed domain wall in the insulating CCDW and the metallic textured CDW states, we focused on the DW-2. However, as pointed by a reviewer, it would be important to verify that our calculations are applicable to reproduce the electronic structures of other domain walls measured by STM and STS. Thus, we added DFT results for DW-4 in the revised supplementary information (see below), which are largely consistent with the case of DW-2.

Comment 2-3 : In the figure 3 caption please specify the energy for (c) and (d).

Reply 2-3 : We appreciate the reviewer’s comment to make our figure captions improved. We added the energy for Fig. 3 (c) and (d) in the Figure and its caption.

Comment 2-4 : When comparing the STS and DFT maps of the domain wall and edge states it appears that they don’t quite match (by rough alignment using Fig. 1b as reference). Specifically, the maxima of the domain wall states (1g) are not located right between the max in 1b, i.e. the centers of the stars-of-David. This might suggest that the reconstruction predicted by DFT is not quite accurate. Please include a figure that shows the position of the domain wall states with respect to the stars-of-

David in the neighboring domains (for the experimental data) and discuss the discrepancy with the DTF results.

Reply 2-4 : We agree the reviewer's comment that there is a remaining discrepancy between experimental and theoretical results for the spatial distribution of the domain edge states. This point was fully discussed for the comment 4 of the other reviewer. In short, we followed the reviewers' suggestion to prepare figures for more detailed comparison of the calculation and the experiment. The partial discrepancies between them are discussed in the revised manuscript.

Comment 2-5 : There are minor wording errors/typos that I assume will be taken care of in the next iteration.

Reply 2-5 : We carefully checked our initial submitted version, again. We found some errors/typos and correct them throughout the manuscript.

Reviewers' comments:

Reviewer #1 (Remarks to the Author):

Re-evaluation of "Correlated electronic states at domain walls of a Mott-charge-density-wave insulator 1T-TaS₂" by Doohee Cho et al.

Overall I believe Doohee Cho et al have addressed the most vital points and proved that there is indeed a band gap around Fermi level with edge states surrounding the gap that they visualized. While I believe the current version with its scientific results and message are ready to publish in nature communication there are some minor points that should be addressed around the theory section:

In the response to the reviewers as well as in the supplementary information the authors show the comparison between the theoretical and experimental spectra. The overlap between experiment and theory is not as good as it sounds from the main text. There are several experimental peaks around the Fermi level that are observed experimentally and are not reflected by any peak in the theoretical calculation, not matter if the theoretical peaks are convoluted by a 2.7 meV Gaussian (equivalent to a 1meV lock in oscillation amplitude at 4K) or a de convolution of the experimental peaks by a 25meV Gaussian to reflect the theoretical peaks. This is especially the case when measuring next to the grain boundary. This is not surprising for such a complicated system. However, referring to the theoretical results to argue "unambiguously" the correlation of these edge states, when there are still some discrepancies in the spectral overlap between theory and experiment and the wave function localization does not overlap at all is simply an overstatement.

I would suggest claiming that the theoretical results strongly suggest that the edge states seem to be a correlated system. Discussing more openly the discrepancies is important, since it creates new questions. Especially the discrepancies in the spatial distribution of the state localization are considerable and right now they are marginalized. I believe important findings such as this will also provide new insight that is not understood and creates the next challenges for investigation.

Reviewer #2 (Remarks to the Author):

The authors have addressed the points I raised during the first review and I now recommend publication of the manuscript in Nature Communication.

Our response to the reviewer's comment.

Reviewer comment:

In the response to the reviewers as well as in the supplementary information the authors show the comparison between the theoretical and experimental spectra. The overlap between experiment and theory is not as good as it sounds from the main text. There are several experimental peaks around the Fermi level that are observed experimentally and are not reflected by any peak in the theoretical calculation, not matter if the theoretical peaks are convoluted by a 2.7 meV Gaussian (equivalent to a 1meV lock in oscillation amplitude at 4K) or a de convolution of the experimental peaks by a 25meV Gaussian to reflect the theoretical peaks. This is especially the case when measuring next to the grain boundary. This is not surprising for such a complicated system. However, referring to the theoretical results to argue “unambiguously” the correlation of these edge states, when there are still some discrepancies in the spectral overlap between theory and experiment and the wave function localization does not overlap at all is simply an overstatement.

I would suggest claiming that the theoretical results strongly suggest that the edge states seem to be a correlated system. Discussing more openly the discrepancies is important, since it creates new questions. Especially the discrepancies in the spatial distribution of the state localization are considerable and right now they are marginalized. I believe important findings such as this will also provide new insight that is not understood and creates the next challenges for investigation.

Our reply:

We fully agree with reviewer on this point. Following the suggestion of the reviewer, we revised the corresponding expression in the final version of the manuscript